

# FACA v1 - Fully Automated Co-Alignment of UAV Point Clouds

Nick Schüßler[1], Jewgenij Torizin[1], Claudia Gunkel[1], Karsten Schütze[2], Lars Tiepolt[3], Dirk Kuhn[1],
Michael Fuchs[1], and Steffen Prüfer[1]

[1]Federal Institute for Geosciences and Natural Resources (BGR), 30655 Hanover, Germany
[2]State Bureau for Environment, Nature Protection and Geology Mecklenburg-Western Pomerania (LUNG), 18273 Güstrow,
Germany
[3]State Agency for Agriculture and the Environment of Central Mecklenburg, 18069 Rostock, Germany

**Correspondence:** Nick Schüßler (nick.schuessler@bgr.de)

**Abstract.** We introduce FACA - Fully Automated Co-Alignment, an open-source software designed to fully automate the workflow for co-aligning point clouds derived from Unmanned Aerial Vehicle (UAV) images using photogrammetry. We developed FACA to efficiently evaluate fieldwork with Unmanned Aerial Vehicles (UAVs) on landslides and coastal dynamics. The software applies to any research requiring comparative precise rapid multi-temporal point cloud generation from UAV

imagery. UAVs are an essential element in most contemporary applied geosciences research toolkits. Typical products of UAV flights are point clouds created with photogrammetry, to measure objects and their change if multi-temporal data exists. Ground Control Points (GCPs) are considered the best method to increase the precision and accuracy of point clouds, but placing and measuring them is not always feasible during fieldwork. Co-alignment leads to the local precise alignment of multiple point clouds without GCPs. FACA uses Agisoft Metashape Pro and the Python standard library. The GPLv3 licensed FACA source

code focuses on extendability, modifiability, and readability. FACA works interchangeably from the command line or a custom graphical user interface. We distribute the software with both usage and installation instructions. Three multi-temporal test datasets are available. We demonstrate the utility and versatility of FACA v1 with a multi-year and -region dataset acquired along Germany's Baltic Sea coast. FACA is in continuous open development.

## 1  Introduction

Local topographical data acquisition has been rapidly simplified by the mainstreaming of Unmanned Aerial Vehicles (UAVs), the cheapening of high-quality camera equipment, and the ongoing development of photogrammetry software (Anderson et al., 2019). These developments have established UAVs as tools for change detection in the applied geosciences (Cook, 2017; Sun et al., 2024) and beyond (e.g., Bendig et al., 2013; Kerle et al., 2019). Structure from Motion (SfM) is the process of deriving 3D information from images (Ullmann, 1979).

It is possible to quantify surface changes based on raster data using the Digital Surface Model (DSM) of Difference (DoD) Method. DoD with raster data is performed by subtracting the values of overlaying cells (Williams, 2012). It is commonly used when working with raster data, e.g., satellite derived data, which lack a point cloud equivalent (Brosens et al., 2022). Raster data generated from UAV flights are a further abstraction of the original imagery and due to their 2.5D nature lack information in





certain areas, e.g., overhangs, compared to point clouds (PCs) (Lague et al., 2013). There are multiple approaches to calculate

the distance between two PCs (Diaz et al., 2024). Global nearest neighbor defines the distance for each point as the one to the nearest point in the other PC. This simple approach has the drawback that its results vary dependent on PC density. Local modeling, such as the advanced Multiscale Model to Model Cloud Comparison (M3C2) (Lague et al., 2013), necessitate parameter selection tailored to each specific application but offer greater robustness.

To get meaningful measurements it is important to align PCs. There are multiple contemporary methods to align PCs from

UAV field campaigns. Direct and indirect georeferencing create products for individual surveys. Indirect georeferencing requires Ground Control Points (GCPs). These need to be well distributed throughout the study area and their location measured (James et al., 2017). Direct georeferencing requires precise location and ideally orientation information for each image (Turner et al., 2014) often only possible with correction technologies, e.g., Real Time Kinematic (RTK) or Post Processing Kinematic (PPK), not available to consumer-grade UAVs, which thus do not reach the same level of precision (Carbonneau and Dietrich,

2016). For multi-temporal usage the RTK base station location needs to be accurately known, as the images taken during flight are only accurate concerning the location of the base station. This raises the same practical problems as with indirect georeferencing and its GCPs.

Cook and Dietze (2019) introduced Co-alignment to address these shortcomings by finding common features between multi-temporal images. Doing so co-alignment sacrifices global accuracy to improve local precision. Co-alignment can be combined

with GCP and or RTK to further increase the precision (Nota et al., 2022). Time-SIFT (Scale Invariant Feature Transform), developed by Feurer and Vinatier (2018), led to co-alignment, but relied on GCPs. Li et al. (2017) established United Bundle Adjustment (UBA) based on the same idea as co-alignment, but considered GCPs necessary for further work with the results in a geographic information system. Multiple studies showed that co-alignment outperforms direct georeferencing when comparing relative precision in a multi-temporal data set (Cook and Dietze, 2019; Blanch et al., 2021; Parente et al., 2021; Nota

et al., 2022).

Automation, parameterization, and optimization of photogrammetry workflows were addressed by previous works: Śledź and Ewertowski (2022) investigated the fine-tuning of Agisoft Metashape parameters under the aspect of a singular flight with GCPs as a validation method. Tinkham and Swayze (2021) analyzed the influence of parameters, especially Depth Map Filtering and Alignment Accuracy, on the ability to detect trees. They found that the highest quality increases the overall

point count and no or mild filtering reproduces trees closest to reality. Blanch et al. (2021) have enhanced and automated co-alignment for fixed cameras in combination with GCPs. They state that their approach is not suitable for UAVs as it requires simultaneous photographs from the same locations. Hendrickx et al. (2018) examined the variation inherent in the proprietary algorithms of Agisoft Photoscan Professional, the predecessor of Metashape Professional. They concluded that the software can generate reproducible outputs with few variations, influenced by the quality settings. Variations are most pronounced in

areas with low image overlap and complex terrain.

With the publication of Fully Automated Co-Alignment (FACA) we introduce a user-friendly, but powerful tool to apply co-alignment fast and reproducible in a convenient framework. Its applications can range from fine-tuning settings to a specific area of study, to applying co-alignment to multiple large areas with robust parameterizations.





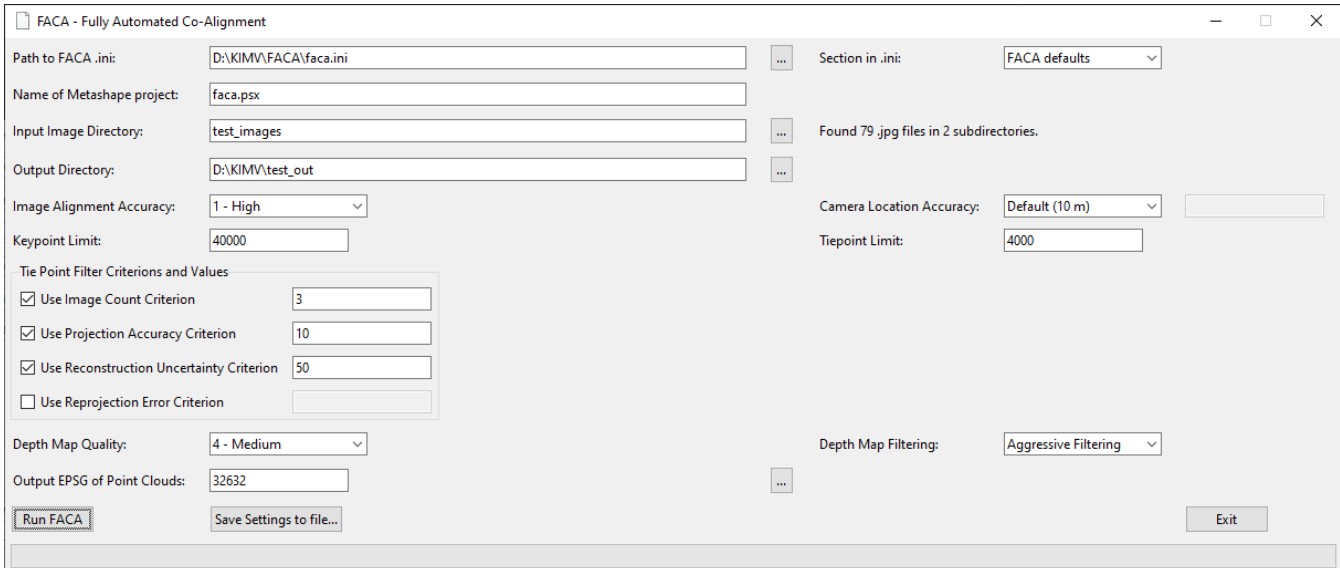

**Figure 1.** The FACA GUI allows the user to load, edit and save parameters and apply co-alignment.

## 2 FACA

FACA is an open-source (GNU General Public License v3) application to automate the co-alignment workflow on a multi-temporal dataset as introduced by Cook and Dietze (2019).

### 2.1 Development motivation

We developed FACA to accelerate the application of the well-established co-alignment method, enabling streamlined deployment at any location with evolving temporal data, minimal preparation and high adaptability and scalability. FACA enhances

both the efficiency and extendability of the co-alignment approach. Our initial goal for FACA was to support our research on the Baltic Sea coast, where we investigate mass movement potentials along cliffs in pilot regions and aim to transfer this knowledge to other areas (Torizin et al., 2024; Schüßler et al., 2023). Data collected during the multiple field campaigns required rapid analysis, interpretation, and dissemination to partners and decision-makers. Additionally, we sought a workflow that could easily be adapted to other coastal sections, and effectively applicable by non-photogrammetry experts. As a result,

FACA emerged as a user-friendly software solution capable of producing reliable results quickly. Although initially developed for UAV imagery, FACA is suitable for aligning large datasets of images, regardless of scale or capture method, provided that adequate computing resources are available. The FACA code was written, commented and licensed to allow for easy modification by other practitioners.





## 2.2 Co-alignment with FACA

FACA, like all variations of co-alignment, leverages computer vision algorithms, that underlie photogrammetry software. These techniques (e.g., Lowe, 2004; Bay et al., 2006; Rublee et al., 2011) detect keypoints — distinctive, invariant locations in an image — and match them across all images. Szeliski (2022) outlines four stages in this mode of operation: detecting, describing, and matching or tracking keypoints. Tracking is an alternative to matching limited to likely keypoints, such as those in similar areas across images. In the case of Agisoft Metashape a proprietary variant (Semyonov, 2011) of the broadly used SIFT algorithm (Lowe, 2004) is employed. Its default matching settings use a-priori knowledge about image locations to preselect images for matching, along with other metadata information, e.g., camera parameters and light conditions. This differentiates the photogrammetry workflow from the core computer vision algorithms, while enhancing performance and precision (Agisoft LLC, 2023a). Images from individual UAV surveys, or generally field campaigns or sessions, are stored in camera groups, and they are all stored in one unit, or 'chunk' in Metashape. Camera groups allow for survey-specific camera parameter optimizations. Tiepoints are keypoints that the algorithm matches across two or more images, establishing spatial relationships when aligned. When tiepoints span images from different surveys they align these surveys together, the eponymous co-alignment. SfM algorithms then use these tiepoints to optimize camera position and orientation through bundle adjustment. Finally, tiepoints are employed in the 3D reconstruction process, generating a sparse point cloud, which can be filtered and refined within the original chunk. Afterwards the surveys are separated, creating individual chunks and discarding keypoints not matched with its images, while preserving location information and camera group specific calibrations. These new sparse point clouds are then processed separately to generate final products, such as point clouds in FACA's case.

FACA introduces flexibility compared to the original co-alignment workflow described by Cook and Dietze (2019), offering several optional steps that users can choose while maintaining the same foundational process. For instance, prior to keypoint matching and alignment, FACA allows users to adjust the accuracy of image location data, which can enhance co-alignment performance in cases where camera location information is uncertain across different surveys. This and other differences stem from the configuration possibilities FACA offers, see Table 1. FACA also differentiates itself from previous co-alignment works by not requiring human intervention, a path forward that Cook and Dietze (2019) already discussed.

## 2.3 Requirements and usage

FACA uses the Python API (Agisoft LLC, 2023a) of Agisoft Metashape Professional version 2.1.0 (Agisoft LLC, 2023b) to automate the co-alignment workflow and is compatible with versions 2.0.0 and later. FACA only requires a Python 3 installation with the standard library, an installed Metashape Python 3 Module, and a licensed copy of Agisoft Metashape Professional.

Every co-alignment relevant parameter that can be modified in the Metashape interface can also be altered via the Python API (Agisoft LLC, 2023a). Metashape also provides the ability to process batches, created using its main interface. This batch process is not a full-fledged replacement for the Python API. In comparison it lacks many features, e.g., modifying existing chunks and labeling objects, in addition to losing the flexibility provided by Python, e.g., a custom interface and to dynamically





name outputs. With the Metashape Java API (Agisoft LLC, 2024) there exists a third option on par with the Python API. We use Python because of our familiarity with the language and its popularity for Metashape scripting.

FACA imposes minimal rules on the input data and its structure, ensuring a smooth, automated workflow while maintaining adaptability for various use cases. The input directory must contain at least two subdirectories, each with one surveys images

as jpg-files, allowing for any number of subdirectories as required by the user, without limiting the complexity or organization of the data hierarchy. All images must contain location information in their metadata. The name of the first level subdirectories will be used to reference the images they contain, as well as the outputs generated from those images. We found it convenient to name them based on their survey date.

FACA allows the use of predefined parameters, which are stored as sections within a configuration file. The default con-

figuration file included with FACA contains a section that defines overarching parameters, such as the image input directory, output directory, and EPSG code for the output point clouds. These default values can be overridden by redefining them in other sections.

There are three ways to interact with FACA. The graphical user interface (GUI) (Fig. 1) uses the tkinter package and is the most accessible, but not as automatable as the alternatives. The interactive mode, where the software asks for each parameter,

while providing the defaults as a suggestion and fallback. The script mode, where the user defines a configuration file and a section therein that FACA uses for the calculation. The user can optionally define parameters that differ from those specified in the configuration file section, for example, to apply established settings to an area with a different EPSG code or to modify input or output directories. If neither a configuration file nor a section is provided, but at least one calculation parameter is specified, the interactive mode will start, omitting prompts for the provided parameters. If all calculation parameters are provided as

arguments, the calculation will start exactly as in script mode. Script mode allows FACA to be part of a larger automated workflow, e.g., applying filtering or color correction on the input images or removing vegetation and calculating statistics on the output PC. The modifiable configuration allows users to easily apply proven co-alignment workflows on their data, as well as find ideal workflows for their study areas with a sensitivity analysis. Configuration files can be created and modified manually or with the help of the GUI. For any other available setting, FACA uses the Agisoft default values, as specified in

Agisoft LLC (2023a).

FACA records its steps and interim results, e.g., the number of detected surveys, corresponding images and tiepoint counts, the Metashape version used, and calculation parameters, in a timestamped log file within the output directory. These small text files can be easily shared with peers to facilitate reproducibility, enhance workflow transparency, monitor performance, and assist in identifying and troubleshooting errors or issues. Additionally, the logs can be automatically parsed to generate figures

for publications or to facilitate the automatic comparison of different parameterizations (Schüßler, 2024).

## 2.4   Parameterization

Parameters and possible values supported by FACA are listed in Table 1. Our streamlined approach allows us to quickly apply multiple co-alignment workflows discussed by other authors and examine their general transferability to other areas. FACA ships with multiple co-alignment parameters used previously in addition to its own defaults ready for use as part of its initial





**Table 1.** FACA parameters and their possible values.

| Parameter | Corresponding Values |
| --- | --- |
| Input Image Directory | Directory with at least two subdirectories, each holding one survey's images |
| Output Directory | Any valid directory |
| Project Name | Any valid filename ending in '.psx' |
| Image Alignment Accuracy | One of:<br>Highest<br>High<br>Medium<br>Low<br>Lowest |
| Camera Location Accuracy | One of:<br>Default (10 m)<br>Exif<br>Custom Accuracy |
| Keypoint Limit | Any non-negative integer |
| Tiepoint Limit | Any non-negative integer |
| Tiepoint Filtering Criterions | None or any of:<br>Image Count<br>Projection Accuracy<br>Reconstruction Uncertainty<br>Reprojection Error |
| Tiepoint Filtering Values | Any non-negative real number |
| Depth Map Quality | One of:<br>Ultra high<br>High<br>Medium<br>Low<br>Lowest |
| Depth Map Filtering | One of:<br>No Filtering<br>Mild Filtering<br>Moderate Filtering<br>Aggressive Filtering |
| Output EPSG Reference | Any valid EPSG code |





configuration file (Cook and Dietze, 2019; Nota et al., 2022; Saponaro et al., 2021; de Haas et al., 2021; Harkema et al., 2023; Omidiji et al., 2023; Moran et al., 2023). FACA emulates the parameters of these co-alignment approaches as far as they do not require additional manual inputs from the user and do not use depreciated features of Agisoft Metashape.

FACA users can choose any settings they want, either from a modifiable list based on previous works or by selecting their own. FACA default parameters emphasize practical automated applications of co-alignment. Because of this, they are 145 conservative, especially the tiepoint filtering, as manual fine-tuning is not expected. We see these parameters as reasonable suggestions and invite practitioners to expand upon them for their needs.

The Image Alignment Accuracy setting controls the level of image scaling applied during camera position estimation (Agisoft LLC, 2023b). Undersampling (Medium, Low, and Lowest Image Alignment Accuracy) the images normally offers faster processing at the cost of detail and accuracy. Oversampling uses more detailed images to provide precise results, but does so 150 at the cost of roughly twice the computing time compared the unmodified images. Using the original image resolution (High Image Alignment Accuracy) is a reasonable compromise.

Camera accuracy defines the expected positional accuracy of each camera's coordinates (Agisoft LLC, 2023b). The default of 10 m maintains some flexibility when using tiepoint matching with posterior knowledge from image metadata. This is crucial, as the accuracy of image location data can vary between UAV flights, often exhibiting different systematic offsets 155 across sessions.

Keypoints are significant features detected in an image, and tiepoints are keypoints matched across images; their limits define the maximal amount of each to consider during processing (Agisoft LLC, 2023b). High or no key- and tiepoint limits can introduce noise into the sparse point cloud and increase processing time, while low values can lead to missing features. We use the default values of 40,000 keypoints and 4,000 tiepoints per image (Agisoft LLC, 2023a), because we found larger 160 values take more time to process with no noticeable quality differences.

Rule-based tiepoint filtering can automatically identify and remove inaccurate or erroneous points, thereby reducing noise, optimizing following bundle adjustments, and improving processing efficiency while enhancing the overall precision in the final product (Agisoft LLC, 2023b). Agisoft Metashape Professional offers four built-in tiepoint filtering criteria: 'Image count' filters tiepoints based on the amount of image they are visible in. 'Reprojection error' uses the maximum difference between 165 measured and parameter adjusted coordinates of a tiepoint, normalized by the scale used. 'Reconstruction uncertainty' uses the ratio of the largest to the smallest semi-axis of the error ellipse of tiepoints. 'Projection Accuracy' uses the average image scale used when getting the tiepoint projection coordinates divided by the number of images containing the tiepoint. We filter tiepoints that are not visible in at least three images, have a reconstruction uncertainty over 50, and projection accuracy over ten.

A depth map is generated for each image, representing the relative distances of objects from the camera's viewpoint (Agisoft LLC, 2023b). The depth map quality settings defines its resolution, and thus the detail and accuracy of the output geometry. 'Ultra high' quality uses the images original resolution and each quality step quartering its pixel count, all the way to 1/256 at lowest depth map quality. We found the medium quality (1/16 resolution) to be suitable for use with UAV imagery.





Filtering the depth map removes tiepoint outliers, and can reduce noise in the output, but smaller details can be lost. In their
documentation Agisoft suggests Aggressive for drone images (Agisoft LLC, 2023b), which is why we employ this setting.

### 2.5  Technical workflow

Fig. 2 shows the processing steps applied when running FACA with $n$ different sessions. Every calculation expects all images
taken during a flight session inside a directory, which, in turn, reside in a common main directory. The lower part of Fig. 2,
below 'Copy Chunk $n$ times', is simplified to show only one individual session $i$, but is applicable for all $i$ from 1 to $n$.
All images are loaded into the original chunk, where each session is organized within its own camera group, to facilitate
individual camera parameter optimization. Each camera group is named according to its survey's subdirectory. Subsequently
the image location accuracy can be optionally adjusted. Options include using the default value of 10 m in each direction
(longitude, latitude, height), setting individual values for each direction, or utilizing precise measurements, e.g., from RTK
equipment, if available. Following this, all images get matched based on the specified alignment accuracy, keypoint limit,
and tiepoint limit. This step aligns all pictures to generate an initial sparse point cloud. The sparse point cloud can then be
optionally thinned using any combination of tiepoint filtering criteria available in Metashape, while optimizing the alignment.
FACA creates a separate copy of the original chunk for each flight survey, removing other session images. Thus, each chunk
contains only the images from the respective session, organized within a single camera group, and a thinned sparse point cloud
that only contains tiepoints visible in that survey. FACA then generates a depth map from these tiepoints with the specified
quality and filtering mode. This depth map is used to construct a dense point cloud. The dense point cloud is subsequently
exported as a las-file in the coordinate system defined by an EPSG code. The name of the final PC output is derived from the
original subdirectory name of the session images.

## 3  Example application

To evaluate its real-world applicability, we use FACA to implement multiple co-alignment workflows in two coastal cliff
erosion scenarios along Germany's Baltic Sea coast.

### 3.1  Study Areas

We applied co-alignment with FACA in two study areas along Germany's Baltic Sea coast, see Fig. 3 and Table 2. The Sellin
study area is located northeast of the town, along the western coastline of Rügen island. Its cliff consists of thick glacial–fluvial
to glacial–limnic sands covering small amounts of till at the foot (Schulz, 1986). The bare, sandy plateau at the cliff top in
the northern quarter contrasts with dense trees in the south. The study area spans approximately 600 meters of coastline, from
coastal kilometer R089.030 to R089.630 (Staatliches Amt für Landwirtschaft und Umwelt Mittleres Mecklenburg, 2010).
The Wustrow study area lies west of Niehagen and Althagen on Fischland, along the western coastline. It is a cliff consisting
predominantly of till, partly overlaid by aeolian sand and cliff-edge dunes (Schulz, 1985). Landwards there is sparse forest in
the southern quarter and agriculture in the north. The coastal retreat is progressing very quickly here due to wind exposure,



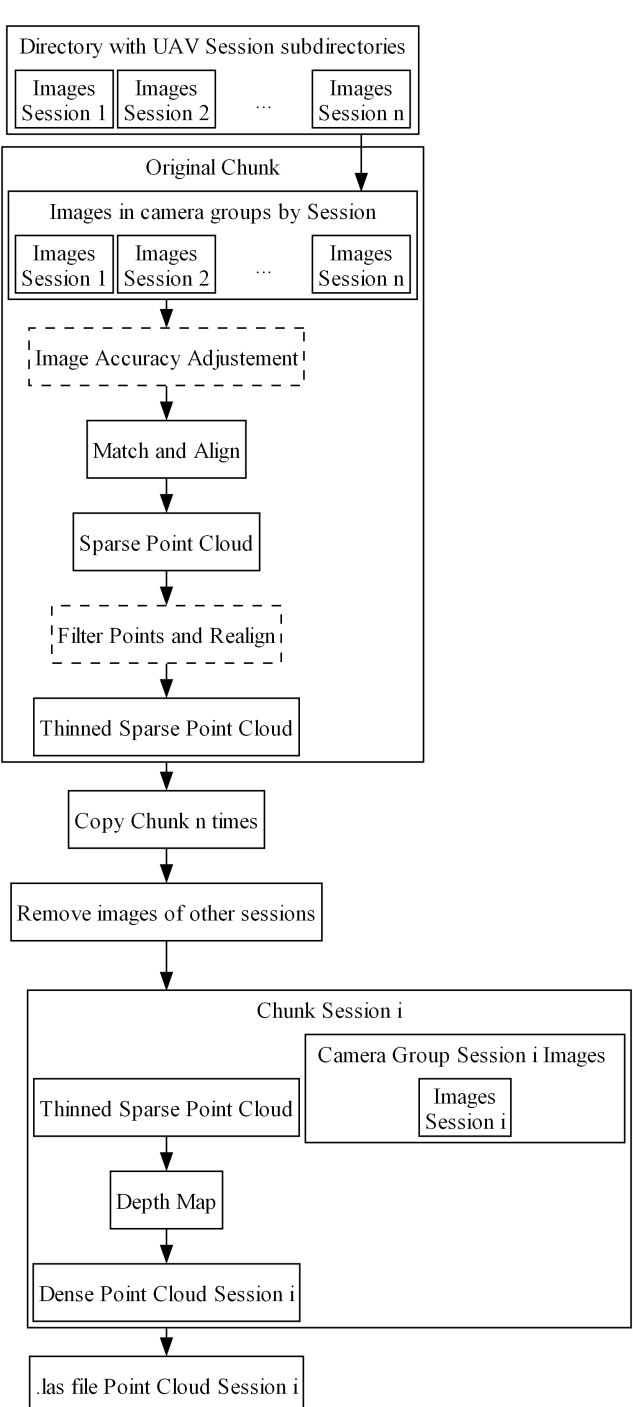

**Figure 2.** FACA workflow. Dashed nodes indicate optional steps.





**Table 2.** Information about study areas and UAV missions.

| Name | Location | Area [m$^2$] | Flight Height [m] |
|---|---|---|---|
| Sellin | 54.38°N, 13.70°E | 52,831 | 40 |
| Wustrow[a] | 54.36°N, 12.39°E | 63,120 | 40 |

a: Four seperate flight missions: 10,526 m$^2$, 12.662 m$^2$, 21,815 m$^2$, 18,117 m$^2$

weak silt material, and its shoreline parallel joint direction (Jacke and Lampe, 2010; Katzung, 2004; Schulz, 1985). North and south of the study area exist coastal protection structures, e.g. groins and breakwaters (Weiss et al., 1983; Bencard, 1998). Attempts to stabilize the coast line in the study area failed in the past (Bencard, 1998). The study area extends over about 1,750 meters of coastline, from coastal kilometer F177.350 to F179.100. (Staatliches Amt für Landwirtschaft und Umwelt Mittleres Mecklenburg, 2010).

Both study areas experience natural coastal erosion and are affected by landslides, leading to rapid changes in the cliff faces, with minimal changes inland.

### 3.2 Data acquisition

We used a DJI Phantom 4 RTK in Terrain Awareness Mode (TAM). The drone is equipped with a 1" CMOS 20 MP image sensor. RTK was enabled for all surveys. We took Nadir and oblique (45°) images along the same flight route. We combined the

official German digital terrain model with 5 m resolution (DGM5) with the German Combined QuasiGeoid 2011 (GCG2011) to program the missions. TAM allows consistent ground sampling distance (1.1 cm/pixel at 40 m above ground for Nadir images) and image overlap. The automated flight plans were created using the DJI GS RTK App that came with the UAV remote controller. We planned for 70% vertical and 80% horizontal overlap of these images. Table 3 shows information concerning the eight field campaigns. Throughout most sessions, we supplemented the automatic pictures with ones taken manually. The

flight time is taken from the metadata of the first and last image taken in the area during that field campaign. In the first field campaign we only flew the southernmost automatic flight mission in Wustrow (10,526 m$^2$). Before field campaign six, in October 2023, we received a second Phantom 4 RTK allowing us to fly parallel sessions. This speeds up the survey time and can reduce lighting differences between photos of the same survey. Starting with field campaign seven in November 2023, we supplemented our automated flights with manual RGB flights conducted using a DJI Mavic 3M. However, we utilized

images from these manual flights exclusively for generating point clouds in the Wustrow bunker area, due to the complex structure following the bunker's collapse. Images of field campaign seven present an additional challenge for our workflow as





**Table 3.** Field Campaign Overview. Empty cells indicate the study area was not surveyed in this field campaign.

| Field Campaign | | Sellin | Wustrow |
|---|---|---|---|
| 1 | Date | 2022-03-15 | 2022-03-16 |
| | Time | 13:52 - 15:32 | 13:30 - 14:08 |
| | Images | 724 | 506 |
| | Usable Images | 722 | 504 |
| | Conditions | Cloudy | Cloudy |
| 2 | Date | 2022-05-17 | 2022-05-18 |
| | Time | 14:44 - 16:01 | 10:27 - 14:19 |
| | Images | 480 | 1,850 |
| | Usable Images | 478 | 1,826 |
| | Conditions | Sunny | Sunny |
| 3 | Date | 2022-11-08 | 2022-11-09 |
| | Time | 12:15 - 13:27 | 10:23 - 12:52 |
| | Images | 481 | 1,848 |
| | Usable Images | 479 | 1,806 |
| | Conditions | Partially Sunny | Partially Sunny |
| 4 | Date | 2023-02-21 | 2023-02-22 |
| | Time | 13:09 - 14:29 | 10:27 - 13:21 |
| | Images | 512 | 2,168 |
| | Usable Images | 465 | 1,997 |
| | Conditions | Sunny | Sunny |
| 5 | Date | 2023-04-18 | 2023-04-19 |
| | Time | 12:53 - 14:18 | 10:40 - 12:11 |
| | Images | 548 | 1,104 |
| | Usable Images | 524 | 1,031 |
| | Conditions | Sunny | Sunny |
| 6 | Date | 2023-10-17 | 2023-10-18 |
| | Time | 13:06 - 13:52 | 10:40 - 12:41 |
| | Images | 505 | 1,997 |
| | Usable Images | 473 | 1,848 |
| | Conditions | Cloudy | Cloudy |
| 7 | Date | | 2023-11-29 |
| | Time | | 11:57 - 13:19 |
| | Images | | 1,845 |
| | Usable Images | | 1,657 |
| | Conditions | | Cloudy, partial snow |
| 8 | Date | 2024-02-29 | 2024-02-28 |
| | Time | 10:25 - 11:11 | 10:27 - 12:31 |
| | Images | 503 | 1,813 |
| | Usable Images | 465 | 1,668 |
| | Conditions | Sunny | Partially Sunny |

the ground is covered by snow, because of this we did not examine Sellin but chose to survey Wustrow, as its steeper cliffs remained exposed.





**Table 4.** Areas with noticeable changes.

| Study area | Description | Location |
| --- | --- | --- |
| Sellin | Sand cliff | 54.3806°N, 13.7031°E |
| Wustrow | Bunker | 54.3611°N, 12.3964°E |
| | Coastal erosion | 54.3642°N, 12.3985°E |

After each field campaign, we reviewed the images and compiled a blacklist of unsatisfactory photos, e.g., poor focus, drone
parts or only sea visible. We excluded these from further processing.

### 3.3 Analysis

Our analysis focuses on the multi-temporal local precision and not the model's global accuracy. We analyzed the local precision
of our results in two ways: Firstly using an area where we saw no change. We used a $60.5 \ \mathrm{m}^2$ section of a roof unaffected by
coastal erosion, in the Sellin study area (54.3794°N, 13.7023°E). There are no suitable anthropogenic structures in the Wustrow
study area, thus we did not measure unaffected areas there. In the stable area, we measured the cloud-to-cloud (C2C) distance
of the PC in the z (height) dimension compared to the first survey. We consider those with the lowest differences best. We
estimated the model's uncertainty with the range of distances in stable areas across surveys. Secondly by examining selected
areas with significant changes, see table 4. The best models would accurately represent the changes we saw in the field in high
detail. The coastal retreat should be recognizable over several flights, without outliers that reverse the trend.

Besides these quality factors, we also measured the processing time, taken from the FACA log-file. We performed every
calculation on the same workstation (CPU: Intel Xeon E-2286G; RAM: 128 GB DDR4 (2667 MHz); GPU: 16 GB Nvidia
RTX 5000; HDD: HGST HUS726T4TALE6L4; OS: Microsoft Windows 10 Enterprise 22H2). The computation times should
be regarded as relative references, not absolutes, as they can vary significantly depending on the available computing resources.

Fig. 4 shows the location of the stable area (pink filled polygon in Fig. 4a), the area used to analyze changes in Sellin (green
polygon in Fig. 4a) and the minimum bounding box of the images used to analyze the changes around the bunker (blue polygon
in Fig. 4b) as well as the one used to quantify the coastal retreat (red polygon in Fig. 4b). The yellow polygons indicate the
minimum bounding box of the automatic images taken during this field campaign. The stable area in Sellin lies just outside the
minimum bounding box of images taken during automatic flight. The unstable areas in Wustrow (bunker in the south, coastal
erosion in the north) are well covered by automatic flight images.

The bunker serves us to visually identify and describe coastal erosion processes, especially after its collapse on the 18th of
February 2024 (between field campaigns 7 and 8).

We used subsets of the Wustrow pictures to analyze the bunker and coastal erosion areas, see Table 5. Both datasets are
freely available under the CC-BY-SA 4.0 license (Schüßler et al., 2024, 2025a). They are the images that were taken within the
defined bounding boxes (blue and red polygon in Fig. 4b) of the unstable areas.

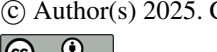


**Table 5.** Number of usable images for the detail calculations of unstable areas in the Wustrow study area.

| Field Campaign | Bunker | Coastal erosion |
| --- | --- | --- |
| 1 | 300 | |
| 2 | 297 | 538 |
| 3 | 297 | 537 |
| 4 | 287 | 555 |
| 5 | 286 | 531 |
| 6 | 309 | 553 |
| 7 | 285 | 489 |
| 8 | 349 | 460 |

We quantified the coastal erosion in Wustrow by tracing the cliff top in the area of interest (see red polygon in Fig. 4b) with CloudCompare (2024) using the generated dense point clouds to create polylines. Then we exported them as Shapefiles and proceeded in QGIS (QGIS Development Team, 2024). In the geographic information system we generated a line grid with the extent set to be slightly larger than the traced cliff tops, 1 m horizontal and 200 m vertical spacing. We discarded all vertical lines and rotated the horizontal ones 20°clockwise, so they are perpendicular to the coastline. Finally, we split the rotated

features with two cliff tops and removed the parts outside. Doing this resulted in roughly 380 measurements of coastal erosion per parameterization. Knowing the days between the UAV surveys we interpolated the yearly coastal erosion. We used the same rotated line grid for all measurements.

    We analyzed the changes in the Sellin cliff by clipping the area of interest (green polygon in Fig. 4a) out of the remaining dense point cloud. The underlying Sellin image dataset (CC-BY-SA 4.0 license) is available (Schüßler et al., 2025b). In this

area, we calculated the M3C2 distance (Lague et al., 2013) for each pair of sequential surveys. We used M3C2 here because it is robust and allows use to measure the direction of changes. We always used the older of the two compared PCs as the basis for sub-sampling to generate core points with 0.25 m spacing. We configured M3C2 to use multi-scale normals between 0.5 m and 4.5 m with 1 m step size and a projection diameter of 0.5 m. The clipping and M3C2 calculation was done with CloudCompare (2024).

FACA automatically logs the timestamps when generating a model. We used these logs to calculate time expenditures and compared them. The best model would both generate fast and offer good local precision.

    For an overview of parameters used in this example application of FACA see Table 6. We chose the parameters discussed in these works because of their range of values and good documentation in their respective articles. This work does not aim to evaluate the results of these previous studies but wants to show the capabilities of FACA across a wide range of parameters.





**Table 6.** Co-alignment parameters used, with their respective publication.

| Parameter Source | de Haas et al. (2021) | Cook and Dietze (2019)[a] | FACA Defaults |
|---|---|---|---|
| Image Alignment Accuracy (Quality) | High | High | High |
| Camera Location Accuracy | 10 m (XY direction) 100 m (Z direction) | 10 m | 10 m |
| Keypoint Limit | 60,000 | 40,000 | 40,000 |
| Tiepoint Limit | 20,000 | 4,000 | 4,000 |
| Tiepoint Filter Criterion(s) (Value) | Image Count (3) | Reconstruction Uncertainty (50) | Image Count (3) |
| | Reconstruction Uncertainty (50) | | Reconstruction Uncertainty (50) |
| | Projection Accuracy (10) | | Projection Accuracy (10) |
| | Reprojection Error (1) | | |
| Depth Map Quality | High | Medium | Medium |
| Depth Map Filtering | Mild | Aggressive | Aggressive |

a: Adjusting the elevation by matching it with the known values from the launch points was not replicated here.

## 3.4 Results

### 3.4.1 Point Cloud generation and filtering

Tables 7 and 8 show the tiepoint counts for the initial and tie- and dense point counts of individual survey chunks. The difference between 'Original' and 'Original - Filtered' are tiepoints removed with filtering methods. There are 153 to 979 remaining tiepoints per image for a survey. The progression of tiepoint counts is shown in Fig. 5 and Fig. 6. These Figures illustrate the number of tiepoints removed by each filtering method. For example, the 'Image Count' criterion, represented by blue bars in both figures, indicates the number of tie points excluded for failing to meet the requirement of appearing in at least three images.

### 3.4.2 Stable Area

Table 9 lists the median cloud-to-cloud distance in the Z dimension (C2C-Z) between each survey and the initial automatic field campaign 1 measured inside the stable area (pink polygon in Fig 4a) and the corresponding dense point count. The arithmetic mean of the values in Table 9, excluding Field Campaign 1, is -0.87 cm for de Haas et al. (2021), 3.5 cm for Cook and Dietze (2019), and 2.8 cm for FACA defaults. Fig. 7 displays violin plots of the C2C-Z between field campaign 1 and each subsequent one. The lines at the end show extrema, while the one in the middle signifies the median value, as listed in Table 9. The range of median distances are 4.3 cm for de Haas et al. (2021), 11.8 cm for Cook and Dietze (2019), and 10.8 cm for FACA default parameters.





**Table 7.** Sellin - tie- and densepoint cloud counts.

| Chunk | de Haas et al. (2021) | | Cook and Dietze (2019) | | FACA Defaults | |
|---|---|---|---|---|---|---|
| | Tiepoints | Densepoints | Tiepoints | Densepoints | Tiepoints | Densepoints |
| Original | 7,522,671 | | 2,415,355 | | 2,417,901 | |
| Original - Filtered | 1,398,923 | | 2,345,909 | | 659,456 | |
| 1 | 340,388 | 107,204,082 | 389,728 | 27,197,422 | 128,273 | 26,035,253 |
| 2 | 127,507 | 88,839,798 | 355,383 | 25,247,471 | 73,299 | 21,832,907 |
| 3 | 193,920 | 78,247,324 | 333,915 | 21,595,366 | 99,943 | 19,935,059 |
| 4 | 167,804 | 84,090,932 | 312,996 | 18,318,959 | 80,519 | 17,886,737 |
| 5 | 161,836 | 72,773,517 | 319,591 | 17,644,617 | 84,436 | 16,876,427 |
| 6 | 207,134 | 126,240,299 | 354,225 | 40,875,624 | 101,190 | 35,936,221 |
| 8 | 215,393 | 72,907,992 | 288,704 | 17,379,256 | 102,155 | 16,623,583 |

**Table 8.** Wustrow - tie- and densepoint cloud counts.

| Chunk | de Haas et al. (2021) | | Cook and Dietze (2019) | | FACA Defaults | |
|---|---|---|---|---|---|---|
| | Tiepoints | Densepoints | Tiepoints | Densepoints | Tiepoints | Densepoints |
| Original | 32,915,985 | | 10,813,529 | | 10,734,079 | |
| Original - Filtered | 6,419,792 | | 10,392,100 | | 3,055,131 | |
| 1 | 265,928 | 107,860,519 | 425,155 | 26,615,198 | 103,955 | 30,801,341 |
| 2 | 962,292 | 624,235,776 | 1,789,144 | 161,735,059 | 484,512 | 158,566,931 |
| 3 | 882,035 | 570,108,945 | 1,589,688 | 129,424,613 | 452,230 | 136,948,711 |
| 4 | 1,378,408 | 596,558,982 | 1,452,323 | 139,890,993 | 560,953 | 144,464,077 |
| 5 | 660,573 | 281,922,140 | 851,614 | 57,242,059 | 293,445 | 54,603,037 |
| 6 | 709,334 | 619,064,595 | 1,305,802 | 144,535,699 | 360,742 | 144,185,327 |
| 7 | 644,905 | 421,392,543 | 1,498,921 | 109,677,766 | 344,893 | 123,933,223 |
| 8 | 931,973 | 548,296,396 | 1,485,176 | 122,626,458 | 469,025 | 132,105,681 |

### 3.4.3 Unstable Areas

Fig. 8 contains point clouds of the sand cliff for each of the surveys flown in Sellin generated with FACA default parameters, clipped to the green polygon in Fig. 4a. Fig. 9 displays the changes between subsequent field campaigns in the Sellin sand cliff as M3C2 distance point clouds. Fig. 10 shows traced cliff tops in Wustrow for Field Campaigns 2 to 8. Statistics quantifying the

interpolated yearly coastal erosion between Field Campaign 2 (16th May 2022) and both Field Campaign 7 (29th November 2023) as well as Field Campaign 2 and Field Campaign 8 (28th February 2024) are listed in Table 10. Fig. 11 contains point





**Table 9.** Sellin stable area - Median Cloud-to-cloud distances in the z dimension (C2C-Z) compared to field campaign 1, and dense point cloud count.

| Field Campaign | de Haas et al. (2021) | | Cook and Dietze (2019) | | FACA Defaults | |
|---|---|---|---|---|---|---|
| | Median C2C-Z [cm] | Points | Median C2C-Z [cm] | Points | Median C2C-Z [cm] | Points |
| 1 | 0 | 42,016 | 0 | 10,467 | 0 | 10,476 |
| 2 | -2.20 | 29,447 | -0.10 | 7,401 | 0.10 | 7,432 |
| 3 | -1.60 | 25,318 | 1.90 | 6,286 | 1.40 | 6,360 |
| 4 | -0.90 | 30,898 | 3.40 | 6,712 | 2.40 | 6,741 |
| 5 | 2.10 | 25,240 | 8.40 | 6,365 | 6.30 | 6,359 |
| 6 | -1.50 | 27,634 | -2.20 | 7,636 | -2.00 | 6,777 |
| 8 | -1.10 | 25,636 | 9.60 | 6,267 | 8.80 | 6,304 |

**Table 10.** Wustrow - Coastal Erosion. Mean, median and standard deviation ($\sigma$) of interpolated yearly coastal retreat.

| | Yearly Coastal Retreat [m a$^{-1}$] | | | | | |
|---|---|---|---|---|---|---|
| | 2022-05-18 - 2023-11-29 | | | 2022-05-18 - 2024-02-28 | | |
| Parameter Source | Mean | Median | $\sigma$ | Mean | Median | $\sigma$ |
| de Haas et al. (2021) | 0.82 | 0.39 | 0.85 | 2.34 | 2.29 | 1.11 |
| Cook and Dietze (2019) | 0.85 | 0.54 | 0.73 | 2.31 | 2.32 | 1.00 |
| FACA Defaults | 0.81 | 0.33 | 0.86 | 2.34 | 2.29 | 1.12 |

clouds of the bunker area (blue polygon in Fig. 4b) in the Wustrow study area for each survey generated with FACA default parameters.

### 3.4.4 Computational Resource Efficiency

Table 11 lists the calculation time for each study area and co-alignment parameterization.

### 3.5 Discussion

### 3.5.1 Point Cloud generation and filtering

The higher key- and tiepoint limits of the de Haas et al. (2021) parameters result in over three times the original tiepoints compared to the other two parameter combinations. The filtering methods were consecutively applied as they are listed in

Table 1 from top to bottom. This results in the first applied filter removing obvious points that most likely would have also been filtered by other methods, e.g., a tiepoint based on just two images also is likely to have a low projection accuracy. The reprojection error filter used with de Haas et al. (2021) parameters removes no points in both study areas, because it is the



**Table 11.** Total and Selected Sub-step Time Expenditure in Hours:Minutes:Seconds.

| Parameter Source | Processing Step | Sellin | Wustrow | Wustrow - Bunker | Wustrow - Cliff |
|---|---|---|---|---|---|
| de Haas et al. (2021) | | | | | |
| | Match and align orig. Chunk | 02:49:55 | 11:21:00 | 01:41:14 | 03:23:04 |
| | Filter orig. Chunk | 00:02:58 | 00:11:25 | 00:01:06 | 00:03:31 |
| | Generate Dense Clouds | 08:08:26 | 36:01:38 | 05:25:33 | 11:26:38 |
| | Export Point Clouds | 00:05:44 | 00:37:34 | 00:04:30 | 00:09:12 |
| | Total | 11:09:01 | 48:21:39 | 07:13:25 | 15:04:30 |
| Cook and Dietze (2019) | | | | | |
| | Match and align orig. Chunk | 01:27:07 | 05:31:43 | 00:58:10 | |
| | Filter orig. Chunk | 00:00:50 | 00:02:16 | 00:00:18 | |
| | Generate Dense Clouds | 02:10:27 | 10:03:17 | 01:28:51 | |
| | Export Point Clouds | 00:01:35 | 00:08:25 | 00:01:14 | |
| | Total | 03:40:40 | 15:53:11 | 02:29:10 | |
| FACA Defaults | | | | | |
| | Match and align orig. Chunk | 01:27:10 | 06:30:38 | 00:59:09 | 01:31:07 |
| | Filter orig. Chunk | 00:01:32 | 00:03:36 | 00:00:31 | 00:00:57 |
| | Generate Dense Clouds | 02:04:09 | 09:35:07 | 01:28:45 | 03:01:24 |
| | Export Point Clouds | 00:01:29 | 00:08:36 | 00:01:14 | 00:02:26 |
| | Total | 03:35:57 | 16:25:13 | 02:30:11 | 04:37:37 |

last applied criterion and the tiepoints have been thoroughly thinned beforehand. Despite this, we consider the application of multiple filters useful, because they apply different quality criteria to the tiepoints and are not time-intensive. The threshold

of each filter should ideally be based on the individual survey flown, but the differing surveys examined in this work show that generalizations are feasible. The FACA default parameters generate the lowest tiepoint count after filtering. The Cook and Dietze (2019) parameterization starts with nearly the same amount of tiepoints, but they remain with over two and three times the de Haas et al. (2021) and FACA default ones after filtering. FACAs conservative filters aim for less noise while accepting the trade-off of a sparser point distribution. This aligns with the FACA goal of being able to work fully automated,

without a human in the loop to validate each point cloud. Given that the method of co-alignment is fundamentally based on finding tiepoints between multiple surveys we consider the image count filter with a threshold $\geq 3$ the most important one, especially given its easy-to-interpret nature. This assures that only tiepoints remain that are detected across multiple surveys and or images in one survey, while none remain that only exist in two images of a or two different surveys. Despite this, the amount of remaining tiepoints per image in each survey and parameterization differs greatly. In addition to different filtering

methods and changes in the landscape between field campaigns and or study areas, we interpret part of the fluctuation as a result





of inefficient manual image capturing. This highlights the efficiency of well-planned automatic surveys, to at least generate a general baseline to optionally expand upon with pictures taken by hand.

### 3.5.2 Stable Area

Stable areas between surveys are a prerequisite for the co-alignment workflow (Cook and Dietze, 2019). The comparatively low
global accuracy results in very slight shifts in the position of the polygon selected for comparison of the stable area between different co-alignment parameterizations. It never shifts to unstable areas, i.e., off the roof. Ideally, the median cloud-to-cloud height distance between survey 1 and subsequent ones in stable areas is close to zero, because there were no observable changes in the stable area during our field campaigns, e.g., no leaves on the roof. We explain the differences with uncertainties in the creation of tiepoints, inaccurate camera location information, and the location of the stable area outside the main area
covered by automatic flight routes among others. The de Haas et al. (2021) parameters result in the overall lowest median C2C distances in the height dimension. A contribution to the lower ranges with the de Haas et al. (2021) parameters is that we used the nearest neighbor distance to calculate the distances. de Haas et al. (2021) parameters result in denser point clouds, with between roughly three to four times the points compared to the other two parameterizations, see Table 9. The denser points increase the probability of close correspondence in the global C2C. We did not use a local C2C model or the more advanced
Multiscale Model to Model Cloud Comparison (M3C2) approach (Lague et al., 2013) here because they would ideally require specific settings for each output cloud. Choosing the point cloud of field campaign 1 as a reference is ideal because it is the densest one. Additionally, this choice facilitates the interpretation of changes relative to initial values. Picking the cloud with the highest amount of points as a reference minimizes the overall nearest neighbor distance. We interpret the range of values in the Sellin study area as a result of the flight height and location of the rooftop. The violin plots of Fig. 7 show that the
C2C distance (Z dimension) distribution is similar for different parameterizations. We interpret outliers, e.g., Field Campaign 6 in Fig. 7b, as errors in the calculation of Agisoft Metashape. Overall we show that there are suitable stable areas for co-alignment in the Sellin study area for co-alignment. Calculating the range of the median C2C distance (Z dimension) allows us to better interpret the uncertainties when examining the unstable cliff area in Sellin. While they are too small for a meaningful comparison between parameters, due to the aforementioned low global accuracy, there still exist stable objects in the Wustrow
study area (e.g., benches, bins, and other smaller anthropogenic objects), underpinning our results.

### 3.5.3 Unstable Areas

We decided to limit the difference calculation to the actual sand cliff, but against filtering out the thin vegetation therein as it is part of the moving landslide mass. Overall we observed more pronounced erosion in the northern quarter of the cliff, because of the sparser vegetation in the cliff and at its top. Despite the far southern side not being inside the bounding box of
automated flights, see Fig. 4, there are no visible anomalies in the M3C2 calculation (Fig, 9). Between the first and second field campaigns (Figs. 8a, 8b) there is little change (Fig. 9a), due to the small time delta (63 days). Nonetheless one can already see the movements typical for the cliff: flows in the sand cliff face and slides at the foot of the cliff. They are mutually reinforcing. The slides occur when the Baltic Sea abrasion removes material from the foot of the slope, causing the slope angle to steepen





and the sand to slide down the slope until it reaches its angle of repose, or temporarily steeper if supported by vegetation.
The effects are intensified by precipitation. Alternatively once dried, the sand loses sufficient cohesion to become unstable and begins to flow. There are two topples of vegetation bound material visible in the southern third near the cliff top in Fig. 9b, which shows the difference between Figs. 8b and 8c. In the 105 days separating Figs. 8c and 8d the general trends continue and two larger mass movements are visible in Fig. 9c, one in the southern quarter and slightly north of the center. Fig. 9d shows further flowing of the aforementioned areas. Between Figs. 8d and 8e the southernmost area started to slide and flow
too, while another movement occurred north of the southern flow. Fig. 9e is harder to interpret because of the vegetation growth in roughly half a year. When directly comparing Figs. 8e and 8f we see that the flow slightly north of the center continues, which is not visible in Fig. 9e. Figs. 8f and 8g show the most drastic differences (Fig. 9f) of the cliff yet. We see two large (> 10 m width) slides in the northern part, multiple toppled trees from the crown, and very pronounced movement in the flow regions, that seem to have turned into deeper seated slides. These changes reveal glacial till beneath the sand, which leads to
the cliff steepening beyond the sand's angle of repose. Once the coastal erosion weakened the glacial till it could lead to larger single-event landslides in that area.

We did not use the PCs generated for the whole Wustrow study area created with de Haas et al. (2021) and FACA Default parameters, due to alignment problems most likely caused by the movement of the RTK base station. Instead, we generated new, smaller PCs to analyze coastal erosion, employing the same parameters. The values in Table 10 were interpolated based
on the cliff top's position in the second, seventh, and eighth field campaigns, as the first survey did not cover this area. There are inherent uncertainties when tracing the cliff top by hand to measure coastal erosion. The variation between yearly coastal retreat values for different co-alignment parameterizations is most likely attributable to these uncertainties. However, all generated PCs were sufficiently dense and detailed to allow accurate tracing of the cliff crown, with no uncertainty regarding its location. The difference in cliff top measurements in the seventh and eighth field campaigns in Fig. 10a indicate that the erosion process is
highly inconsistent, likely driven by the significant impact of storm surges and severe weather events. The Figs. 10b-d further demonstrate that the retreat is not uniform along the coastal cliff, with retreat occurring more rapidly in zones of weakness, eventually isolating and eroding more stable areas. Overall the measured coastal retreat between survey two and seven is consistent with previous estimates for this area, which range from 0.8 to 1.4 $\mathrm{m\,a^{-1}}$ as collected by Schulz (1985) and from 0.38 to 1.09 $\mathrm{m\,a^{-1}}$ by Ullrich (1983). The calculations by Hoffmann and Foy (2023) of 0.65 $\mathrm{m\,a^{-1}}$ (1835 - 2021) and 0.74
$\mathrm{m\,a^{-1}}$ (2005 - 2021) are both close, but slightly lower than, our results. The over 2.3 $\mathrm{m\,a^{-1}}$ coastal decline we measured between survey two and eight is an outlier compared to previous results. We attribute this discrepancy to the impact of a recent storm surge and the relatively short time span of just under two years. Additional drone surveys could provide more data and reveal a clearer trend. This would be particularly valuable, as older documents and maps, such as those from the 19th century, which are currently used for long time difference calculations (Hoffmann and Foy, 2023; Schulz, 1985), often are
hard to georeference, leading to significant uncertainties. New data would also be valuable for investigating and monitoring the assumed impacts of climate change (Meier et al., 2022).

The progress of coastal erosion in the southern part of the Wustrow study area is visualized in Fig. 11. The pictured bunker is located approximately 350 m beyond the southern end of the traced cliff top (Fig. 10), see Fig. 4b. During its construction





for the East German army, the building was multiple meters inland. The loose material with partially intact vegetation at the
bottom of 11a indicates that there has been a recent landslide. A coastal retreat occurred between Figs. 11a and 11b (north of
the landslide debris seen in 11a). We interpret it as the result of abrasion from the feet of the slope by the Baltic Sea and thus its
steepening in combination with a loss of cohesion inside the material and change of geometry due to the older landslide at its
edge. Additionally Fig. 11b shows the removal of loose landslide debris at the foot of the cliff north of the bunker. This trend
continues in Figs. 11c and 11d, revealing a boulder that is heavy enough to defy tidal forces. Fig. 11d first shows the coastline
retreating far enough along the borders of the bunker to reveal its northern wall. The complete removal of loose material from
the landslide reveals underlying gray glacial till in which a wave-cut notch begins to form (Fig. 11d). This wave-cut notch
widens in Fig. 11e, while the aftermath of an earth fall or topple south of the bunker is visible. Over the summer of 2023, the
erosion along the edge of the bunker moves further inland (Figs. 11e and 11f). Fig. 11f shows the collapse of the wave-cut
notch, the further movement of the cliff top at the sides of the bunker, and over-steepening. Between Figs. 11e and 11f and
especially between Figs. 11f and 11g a steepening of the lower parts of the cliff is visible. These direct abrasion results appear
most pronounced underneath the front of the bunker between 11f and 11g. The erosion at the feet and sides of the bunker in
combination with its high specific weight and inflexibility resulted in its tople, see Fig. 11h. Throughout the time series, the
effects of precipitation-induced erosion are visible at the bunker walls facing the Baltic Sea. The PCs in Fig. 11 highlight the
local luv-lee effect, where forces are diverted from the robust bunker into the susceptible material surrounding it. Given the high
rates of coastal erosion in the area and the lack of coastal protection, it was only a question of time before the bunker toppled
into the Baltic Sea. We perceive that there were signs of a potentially imminent bunker topple as early as November 2023. The
erosion has progressed far enough underneath the bunker that the dense point cloud contains holes (Fig. 11g) pointing to very
steep slope angles only possible due to the robust bunker. These circumstances combined with a storm surge (Perlet-Markus,
2023) resulted in the final toppling of the bunker (Fig. 11h). This event highlights the potential dangers posed by dynamic
coastal processes and underscores the importance of using such occurrences to raise public awareness of natural hazards.

### 3.5.4 Computational Resource Efficiency

As expected the matching and aligning of the original chunk takes nearly the same time in each area, for Cook and Dietze
(2019) and the FACA defaults because they use the same parameters for this step. The additional hour required with FACA
defaults, compared to the parameters from Cook and Dietze (2019), to match and align the Wustrow study area is attributed
to background usage of the workstation, not differences in parameterization. Otherwise, the values in Table 11 align with our
expectations. Compared to the total time taken to apply the workflow, the time expenditure to filter the tiepoints is negligible,
yet we still see differences based on the tiepoint count and the number of filtering methods applied. Time taken to filter out
tiepoints might even accelerate the latter steps in the workflow because there are fewer points to consider when generating
dense point clouds. When generating dense clouds we observed similar processing times for both Cook and Dietze (2019) and
FACA defaults parameters. In contrast de Haas et al. (2021) took around four times longer. This stems from the higher keypoint
limit and high-quality depth maps the latter uses. We noticed no significant quality differences in the final dense clouds. This
is evident by the similar results across parameterizations in Table 10. Other researchers might benefit from the slightly lower



uncertainties in stable areas, as shown in Fig. 7 and Table 9, and may consider the longer computation time an acceptable trade-off. The amount of unfiltered tiepoints scales nearly linearly with the matching alignment time. It takes not even an hour

(< 3 %) longer to apply the whole workflow on every study area with both Cook and Dietze (2019) and FACA defaults than it takes to just apply the workflow in Wustrow with de Haas et al. (2021) parameters. This shows that the chosen parameters have a sizable impact on the necessary computing and time expenditure.

## 4 Conclusions and outlook

Co-alignment is a good technique to prepare point clouds for change detection and time series analysis. We showed that it

provides good results even if the terrain has undergone extensive, study area-wide change, e.g., after snowfall, given enough alternative surveys to generate common tiepoints. Our comparative study of different co-alignment parameters shows that FACA is an excellent tool to easily generate reproducible research. We hope that the streamlined approach to co-alignment offered by FACA allows researchers to work around some of the method's drawbacks, i.e., the impact of time-consuming recalculations after new surveys can be lessened by the automation. The easy configuration allows practitioners to find suitable

parameters for their study area. Detailed logging can grant insights into the workflow.

Moving the workflow beyond Metashape either as a stand-alone solution or to free software, e.g., OpenDroneMap Authors (2020), could be a next step. Doing so would significantly lower the financial barrier to entry, just as FACA lowers it for configuring and using co-alignment. The more open approach would also allow for co-alignment specific optimizations, that are usually not viable in default photogrammetry workflows, such as defining different quality criteria for tiepoints based on

whether the images were taken during the same survey or not.

Modifying FACA to allow the usage of GCPs would further improve the software. Using computer vision this task could potentially also be automated. Another interesting step forward would be to use a-priori knowledge to designate or automatically detect (Peppa et al., 2018) stable areas to establish ground control areas, to minimize the noise in that area, and thus, better align every part of the study area. This could be done on the stable subset with the iterative closest point algorithm and

the other parts of the point cloud are then transformed to fit the registered stable areas. The custom software approach would also benefit this idea, as tiepoints in these areas could be treated differently from the ones in dynamic areas.

We hope that the methods introduced in this article help widen the adoption of UAVs for multi-temporal dataset generation, not only in the field of natural hazards but also in other parts of the applied geosciences, e.g., geotechnics, geomorphology, and opencast mining.

We publish our code as open-source software to encourage not only its use but also to allow others to examine its underlying mechanisms, contribute their ideas, and foster discussion among users.

*Code and data availability.* The latest version of FACA is freely available at https://github.com/BGR-EGHA/FACA under the GPL v3 license. We archived the version used create our results on Zenodo: BGR - Engineering Geological Hazard Assessment (2024). Schüßler





(2024) contains scripts for data visualization and result analysis. The bunker images are provided by Schüßler et al. (2024), while Schüßler et al. (2025a) contains images of the area where we measured changes due to coastal erosion. Additionally, the Sellin images are available from Schüßler et al. (2025b).

*Author contributions.* All authors participated in the fieldwork. NSC developed FACA and wrote this paper with input from all authors. JT helped develop FACA, contributed to the paper, and managed the underlying project. KS provided local geological knowledge and insights about coastal dynamics. LT supplied information about coastal management and processes. MF contributed with knowledge about UAV usage, local geology, and testing the software. CG extensively tested the software. DK, and SP provided valuable input and helped write the paper.

*Competing interests.* The authors declare that they have no conflict of interest.

*Acknowledgements.* The manuscript was proofread with the help of AI tools. The authors gratefully acknowledge the contributions of the following colleagues to the successful field work: Kai Hahne and Patrick Reschke.



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







**Figure 3.** Study areas overview. (a) Location of Mecklenburg-Western Pomerania in Germany; (b) location of the study areas in Mecklenburg-Western Pomerania; (c) northern part of the Sellin study area (2024-02-29); (d) southern part of the Sellin study area (2024-02-29); (e) central part of the Wustrow study area (2023-10-18); (f) bunker in the Wustrow study area pre-failure (2022-03-16); (g) the bunker post-failure (2024-02-28).



**Figure 4.** Orthophotos based on images from field campaign 6 with stable (pink filled polygon), sand cliff (green polygon), bunker area (blue polygon), coastal erosion area (red polygon) and the minimum bounding box (yellow polygons) of all images taken automatically on that day. (a) Sellin; (b) Wustrow.



**Figure 5.** Sellin - Filtered Tiepoints by method and remaining ones.



**Figure 6.** Wustrow - Filtered Tiepoints by method and remaining ones.



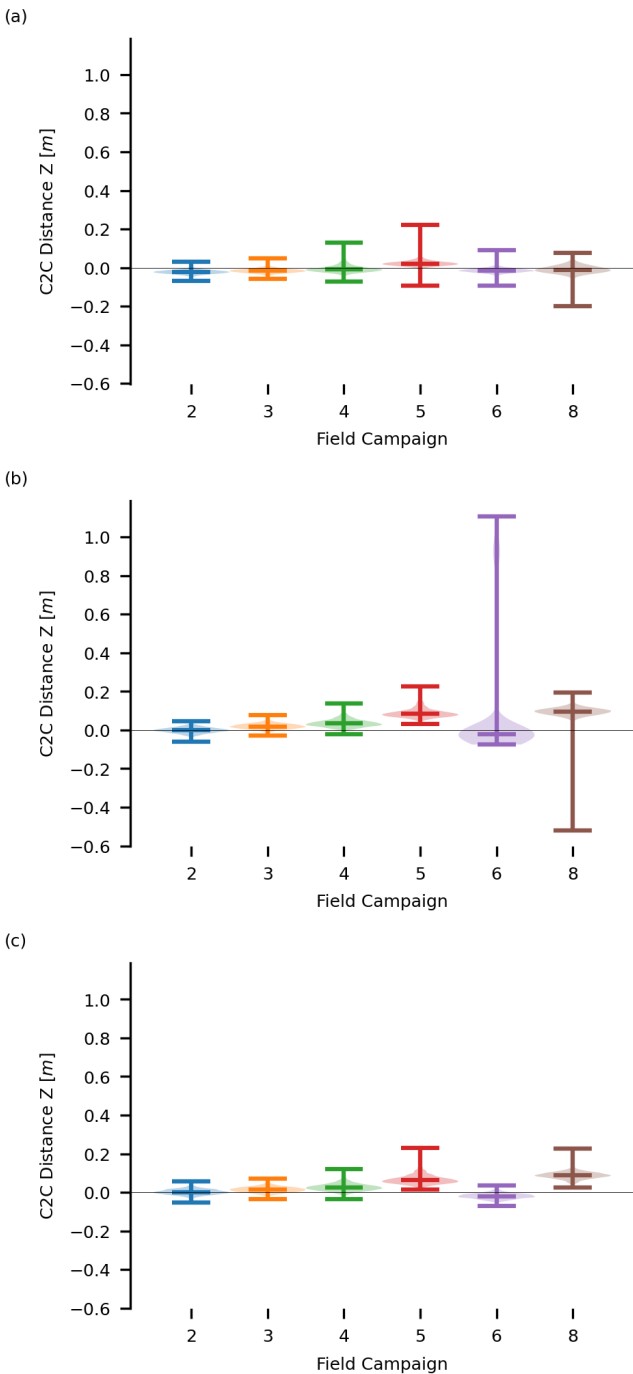

**Figure 7.** Sellin stable area - Violin plots of the cloud-to-cloud distances (Z dimension) with field campaign 1 as reference. (a) de Haas et al. (2021) Parameters; (b) Cook and Dietze (2019) Parameters; (c) FACA Default Parameters.





**Figure 8.** Dense Point Clouds showing the cliff in the Sellin study area of field campaigns 1 to 6 and 8 from the North-East based on FACA default parameters.







**Figure 9.** Dense Point Clouds showing the M3C2 distance calculated based on FACA default parameters of the cliff in the Sellin study area of field campaigns 1 to 6 and 8 from the North-East.





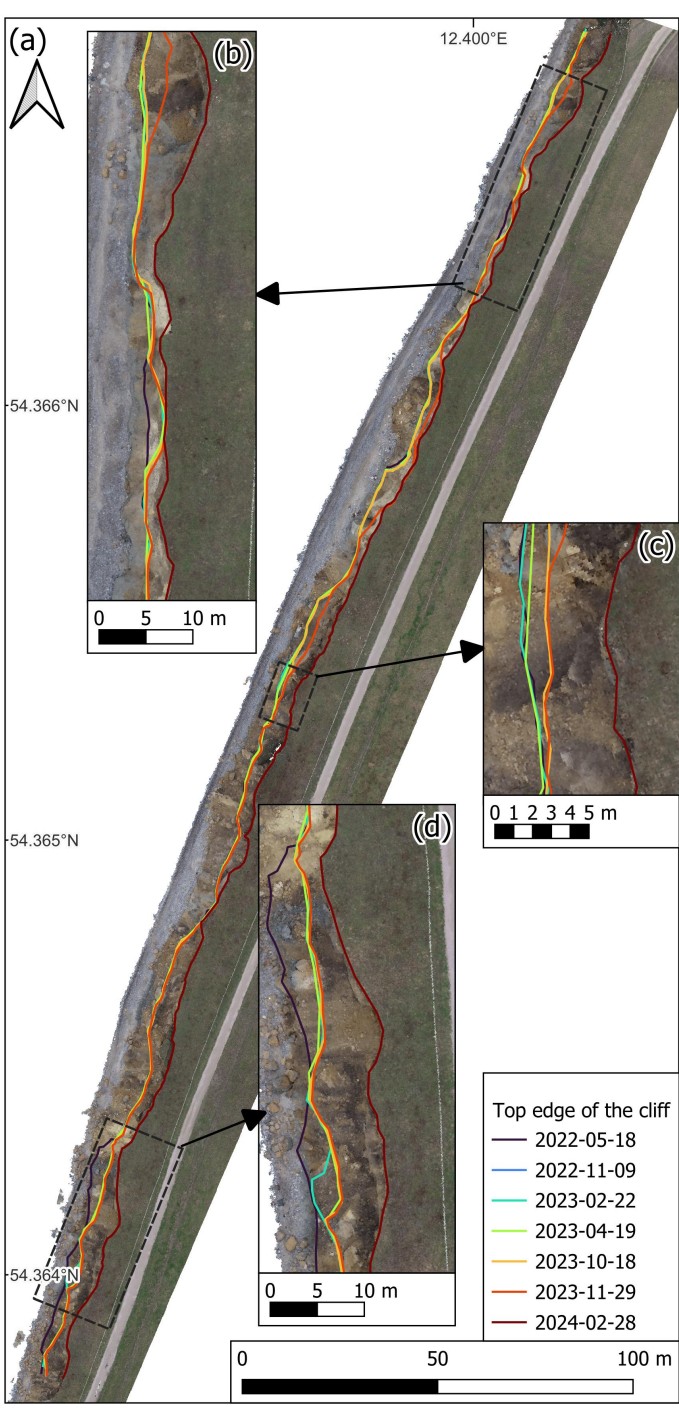

**Figure 10.** Cliff top development in Wustrow. (a) Top edge of the cliff from point clouds of field campaigns two to eight with an orthophoto generated from the 2024-02-28 Flight with FACA Default Parameters; (b) northern detail; (c) center detail; (d) southern detail.





**Figure 11.** Point Clouds of the Bunker in the Wustrow study area from the northwest, generated with FACA Default parameters for field campaigns 1 to 8.