# Peer review of "FACA v1 - Fully Automated Co-Alignment of UAV Point Clouds"

_Geoscientific Model Development, 2024_

## Author Response (AR1)

**Author's response**

**Response RC1**

Dear Ms. Becky Collins,

thanks for your valuable comments and remarks.

Please see our detailed responses to your individual points below:

> The introduction of structure from motion would benefit from a couple of additional sentences of detail to provide more background on the technique and to give context for the subsequent work.

We have expanded the introduction to clarify the use of structure from motion (SfM).

> I think the introduction should contain quantitative critique of the existing methods. There is very little comparison between Direct and Indirect methods for georeferencing or the difference between the accuracy of a point cloud generated from GCP aligned data and that making use of a co-aligned or indirect method of alignment. For a newcomer to the field who may be reading this, it might be helpful for you to define the differences in accuracy between different existing techniques to highlight expected error. This could be done effectively with a simple table of typical errors of the various techniques.

While a detailed quantitative critique of direct and indirect georeferencing is beyond the scope of this manuscript, we acknowledge that the differences between these approaches were not sufficiently outlined in the introduction.

We have revised the relevant section to better clarify their distinctions and rationale, and we have incorporated quantitative data from a previous study to support the description of co-alignment.

> Line 219 you mention supplementing the UAV collected data with manually taken images – would be good to know the reasoning for this and what decision making was taken in the field to decide when and where to do this.

We have provided reasoning for when we decided to supplement the imagery in an additional sentence.

> Tables 2 and 3 could be combined together into one. Table 2 as it currently stands seems superfluous.

We have merged Table 2 into Table 3 and updated the corresponding text references accordingly.

> Section 3.3 you discuss the use of C2C distance for comparison of stable areas but earlier you highlighted the problem of C2C distance often being a function of point density. Some additional justification for that choice would be valuable here. Section 3.4 you present the use of the M3C2 method to compare distance to the first field campaign. Again, explanation of why this choice was made vs the already established C2C method would be valuable.

We have added explanations in Sections 3.3 and 3.4 to justify our choice of the C2C and M3C2 methods, respectively.

> Frequent use of very high precision in quoting numbers of points and seconds of analysis. Consider whether this is needed as it can mask the magnitude of the differences between techniques. Perhaps consider quoting points to the nearest hundred or thousand and times to the nearest minute.

We agree on both counts.

We have rounded the quoted points in the tables to increase the readability and to highlight relevant differences, especially given the inherent non-determinism of the software.

Processing times have been rounded to the nearest minute to enhance clarity.

> Table 11 it is clear that the filtering of original chunk and the exporting of the point clouds represent only a very small proportion of the overall time. Excluding them from this table would allow more focus to be given to the components that have a significant impact on the processing time.

We respectfully disagree that the filtering and exporting time expenditure should be excluded, but we agree that the table did not emphasize the key findings.

To improve readability and focus attention on the main message, we have highlighted the "Total" rows in bold.

Retaining the time to filter the original chunk is important because it shows how the small absolute time investment can have an impact on subsequent steps in the workflow and the exported point cloud.

Similarly, the time to export the point clouds correlates with size of the dense point clouds and thus can be useful for the interested reader.

Both rows illustrate relevant relative differences between different possible parameterizations, aiding readers in selecting configurations best suited to their needs.

As stated in the manuscript, we focus on relative differences between parameterizations when looking at time expenditure, acknowledging that available computational resources may vary widely across users.

> The discussion section contains a number of very long paragraphs that could be broken down into smaller sections to better elucidate the points. Breaking the discussion more clearly into sections that discuss the erosion recorded and the accuracy/evaluation of the FACA method would be clearer, especially as this is a paper presenting a methodology and not a paper describing the erosion processes.

To help readability, we have separated the "Unstable Areas" subsubsection into three different ones, adding line breaks to clearly separate distinct points.

Furthermore, we have reduced the emphasis on the erosion processes by rewriting, shortening, and removing some of the more detailed descriptions, particularly in the sand cliff and bunker parts of the discussion.

We do not intend to expand further on the co-alignment method within the example application, as its primary purpose is to demonstrate exemplary workflows integrating FACA.

The focus of the manuscript remains on the methodology, with the example serving to illustrate the broad applicability of co-alignment with FACA.

> There could also be some attention paid to the concision of the discussion – for example, line 370-371 contain a short sentence telling of the uncertainty of tracing cliff tops and the next sentence then attributes variation in coastal retreat to these uncertainties. However this could be achieved by rewriting to something like – "The variation between yearly coastal retreat values for different co-alignment parameterizations is most likely attributable to uncertainties related to the process of hand tracing the cliff top as a measure of coastal erosion."

We have streamlined the discussion, including the specific sentences referenced, to improve conciseness and clarity.

> More detail related to the statement made at line 367 regarding movement of the RTK base station is required. What were the indicators that led you to that conclusion?

We have concluded that relocating the RTK base station mid-session impacted image accuracy because image positions are referenced relative to the base station, and the absolute coordinates of the base station's location were not consistently known.

We have updated the manuscript to clarify this point.

> Captions of figures 5 and 6 need additional information to make it clearer what they represent.

We have updated the captions of figure 5 and 6 to include additional information.

**Response RC2**

Dear Anonymous Referee,

thanks for your valuable comments and remarks.

Please see our detailed responses to your individual points below:

> I do not have any major issues with the article, however I found the logical arguments at the start of the introduction difficult to follow. I think this could have clearer narrative structure and be explained a little better. Also regarding the introduction I was missing an overview of existing software or alternative options and what this software adds to that ecosystem.

We have revised the introduction to introduce other software solutions aimed at aligning point clouds, clarifying their respective aims and positioning FACA within this ecosystem.

> The description in the results in section 3.5.3 is OK, but I'm unconvinced it needs to be so long if the paper aims to introduce the tool. If there are other useful findings and significant points from this analysis perhaps they can be signposted at the end of the introduction as a contribution of the paper.

We agree that the primary focus should remain on FACA.

Accordingly, we have shortened and subdivided the originally verbose discussion 'Unstable Areas' subsubsection.

However, we prefer to keep the example application separate from the rest of the manuscript.

> Line 16 "Structure from Motion (SfM) is the process of…" This sentence hangs without any context or explanation. Can you rewrite the point being made here?

We have expanded the manuscript to explain Structure from Motion and its significance for UAV Point Clouds.

> There is then a big jump in logic to discussing DEM differencing, which is confusing to me and I think the logical structure of the second paragraph could be better.

We agree that the transition lacked clarity.

To improve the logical flow, we have added a sentence explaining the need for DEM differencing and other approaches to examining changes with multi-temporal datasets.

> Line 56: Are other open source software available to accomplish similar tasks or do they all lack some functionality? Some mention or discussion of this would be useful before introducing your tool.

FACA is the first software specifically developed for the co-alignment workflow.

We have updated the introduction with a selection of other alignment related methods to show what is new with FACA.

> Figure 1 needs to be introduced in the text before being presented.

We have repositioned Figure 1 later in the text.

> Around line 145: I was wondering if the description of what each parameter does would be better presented in table 1? Or could a supplementary table be produced with both the parameter, its corresponding values and a simple description of its purpose included. This is by no means a requirement but it would be good to know it's been considered.

We do not believe the parameter descriptions should be included in Table 1, as it would duplicate the information from subsection 2.4.

While we have considered a supplementary table for parameter description, we find that the manuscript profits from the motivation and reasoning behind each parameter in addition to its description.

In our opinion, this combined motivation for the default values and description works best in free text while a table is overly condensed.

> Line 235: How representative is the location used to test the local precision? There is not much discussion of this point, but it seems like there is a rather limited set of objects and conditions with which to assess the alignment with the test data. Is it possible to strengthen the justification of the test sites and identify any limitations? I think these are mentied in 3.5.2 but if they are known a priori it would be good to be upfront.

We have added an explanation why we chose the roof, and mention its drawbacks in the analysis subsection.

> Section 3.5.3 – there is a long description of the features in the imagery, but it's not always clear that this is necessary to present the modelling software. I would be tempted to shorten this but there is nothing wrong with the section if the authors can justify the inclusion.

We agree that the imagery description was overly detailed, and have revised the section to be more concise.

> Line 373 – I assume no uncertainty should be minimal or insignificant uncertainty.

You are correct, and we have updated the manuscript.

> Line 413: "The additional hour required with FACA defaults, compared to the parameters from Cook and Dietze (2019), to match and align the Wustrow study area is attributed to background usage of the workstation, not differences in parameterization." Could this not be controlled for better? It seems an unnecessary lack of precision in the results.

We have rerun the calculation for the Wustrow study area with the FACA default parameters.

The corresponding manuscript text, figures, and tables have been updated to reflect the new values.

**List of all relevant changes made in the manuscript**

The following list summarizes all relevant changes made in the manuscript since the last submission:

- The introduction to Structure from motion has been reworded for clarity
- A new table 1 presenting exemplary mean absolute and relative errors of direct, indirect, co-alignment and co-alignment with GCPs workflows has been added to the introduction
- Reasoning for supplementary images has been added in the data acquisition subsection
- The former Table 2 and Table 3 have been merged into a single, revised Table 3
- The rationale behind the use of each distance calculation method has been expanded
- The values in the Tables 7, 8, 9, and 11 have been rounded
- The old subsubsection 3.5.3 has been divided into the new subsubsections 3.5.3, 3.5.4, and 3.5.5 , and the content has been shortened
- An explanation of the error caused by moving the RTK base station has been added
- The captions of Figures 5 and 6 have been extended for clarity
- A list of comparable software tools, including a discussion of their differences from FACA, has been added to the introduction
- Figure 1 has been repositioned within the manuscript
- A justification for selecting the roof in Sellin as a stable reference area has been added in subsection 3.3
- The Wustrow calculation with FACA default parameters has been rerun and the corresponding subsubsection 3.5.6, Figure 6, and Tables 8 and 11 updated to reflect the new values